# Education-related inequality in overweight and obesity among Mongolian adults

**Munkhjargal Dorjravdan**[1,2]*, **Javkhlanbayar Dorjdagva**[3,4], **Katsuyasu Kouda**[5], **Enkhjargal Batbaatar**[6], **Naranzul Dambaa**[1], **Tsolmon Boldoo**[7], **Bolor-Erdene Ganbold**[8], **Kumiko Ohara**[5], **Chikako Nakama**[5], **Toshimasa Nishiyama**[5]

**1** Tuberculosis Surveillance and Research Department, National Center for Communicable Diseases, Ulaanbaatar, Mongolia, **2** Oddariya Foundation, Ulaanbaatar, Mongolia, **3** Institute of Public Health and Clinical Nutrition, Faculty of Health Sciences, University of Eastern Finland, Kuopio, Finland, **4** Department of Public Health and Health Policy, Hiroshima University, Hiroshima, Japan, **5** Department of Hygiene and Public Health, Kansai Medical University, Hirakata, Osaka, Japan, **6** Department of Social Sciences, Faculty of Social Sciences and Business Studies, University of Eastern Finland, Kuopio, Finland, **7** Country Office Mongolia, World Health Organization, Ulaanbaatar, Mongolia, **8** School of Public Health, Mongolian National University of Medical Science, Ulaanbaatar, Mongolia

* muuji83@gmail.com

## Abstract

The increasing prevalence of overweight and obesity is a serious public health problem in low- and middle-income countries, closely linked to socioeconomic status, with significant disparities observed across different socioeconomic groups. However, the association between socioeconomic status and overweight or obesity has been less studied in Mongolia. The aim of this study was to examine socioeconomic inequality in overweight and obesity among adults in Mongolia. The data for this study was derived from the Mongolian National Tuberculosis Prevalence Survey, which included 41,777 participants aged 18 years and older. We used Erreyger's concentration index to assess the degree of socioeconomic inequality in overweight and obese individuals. Socioeconomic status was measured by educational level. A decomposition analysis was applied to identify the factors contributing to inequality in overweight and obesity among Mongolian adults. Among the study population, 33.4% were overweight, and 20.7% were obese. The Erreyger's concentration index for obesity was 0.059 (p < 0.01) among men and -0.047 (p < 0.01) among women. Furthermore, obesity was concentrated among the higher-educated men and lower-educated women. The decomposition analysis results show that education, employment status, and income were the main contributors to education-related inequalities in obesity for both men and women, except the age contribution. Education-related socioeconomic inequalities in overweight and obesity exist among Mongolian adults. Future national strategies for tackling obesity should address inequalities in the root social determinants.

**Data availability statement:** The dataset used for this study cannot be made publicly available because the data is restricted by the National Center for Communicable Disease (NCCD) of Mongolia, the owner of the Mongolian National Tuberculosis Prevalence Survey (MNTPS) (dataset available at https://tubis.nccd.gov.mn); however, it is available from the NCCD upon a reasonable request. Requests should be submitted to the scientific review committee at the National Center for Communicable Disease, Ulaanbaatar, Mongolia (Email: scientificcommittee@nccd.gov.mn).

**Funding:** The author(s) received no specific funding for this work.

**Competing interests:** The authors have declared that no competing interests exist.

## Author summary

Obesity is a growing health challenge worldwide and contributes to serious diseases like heart disease, diabetes, and cancer. In Mongolia, where the population is rapidly transitioning from a traditional lifestyle to a more urban and modern way of living, obesity rates are increasing. However, little is known about how obesity affects people from different social and economic backgrounds in Mongolia. In this study, we analyzed data from a National Tuberculosis Prevalence survey conducted in 2015 to examine socioeconomic inequality in overweight and obesity among adults in Mongolia.

We found that women with lower education levels are more likely to be overweight or obese, while the opposite is true for men. Understanding social inequalities in overweight and obesity is crucial to develop and implement future public health policies to tackle the burden of obesity in the country.

## Introduction

Overweight and obesity are global public health threats and challenges that are responsible for at least 2.8 million deaths each year [1]. In 2022, 43% of adults globally were overweight, and 16% were obese. The prevalence of overweight and obesity doubled between 1990 and 2022, particularly in low- and middle-income countries (LMICs) [2]. The increasing obesity rates place a substantial burden on healthcare systems and economies, making obesity reduction a public health priority in many countries, including Mongolia.

Mongolia is a middle-income country, located in central Asia, with a population of 3.4 million in 2023, where the education level is high with over 95% of men and women who attended primary school [3,4]. Over the last three decades, Mongolia has undergone rapid economic growth, with its Gross Domestic Product (GDP) per capita tripling from $4,983 in 1990 to $12,275 in 2023, due to foreign investments in the mining sector [5,6]. This economic development has been accompanied by rapid urbanization, with 70% of the population now residing in urban areas, and a shift from a traditional nomadic lifestyle to a more Westernized, sedentary way of living [7–9].

Previous studies have highlighted an excessive consumption of meat and refined grains, inadequate intake of fruits and vegetables, and dietary imbalances exacerbated by national food supply challenges. Additionally, the physical activity level has declined notably among both men and women [10,11]. For example, a study in Mongolia found that the prevalence of low physical activity increased from 8.2% in 2005 to 28.4% in 2019 [12]. Together, these shifts contribute to the rising prevalence of non-communicable diseases (NCDs), including obesity [13].

The prevalence of overweight and obesity in Mongolia has been steadily increasing across all age groups and sexes [14]. Between 2005 and 2019, the prevalence of overweight doubled, increasing from 18.2% to 48.0% among men and from 25.5% to 50.9%

among women. During the same period, the overall prevalence of obesity rose significantly, from 10.2% to 18.5% [9,15,16]. This surge can be partly attributed to rapid urbanization, a dietary shift from traditional foods to a Western-influenced, high-calorie diet in a less regulated food markets, and alongside with decreased physical activity [9,17,18].

Addition to the economic development, the epidemiological transition of diseases was observed between 1990 and 2020, with declining rates of infectious diseases and a rise in deaths from NCDs [13]. The growing burden of NCDs in Mongolia is alarming, with 54.1% of men's deaths and 60.4% of women's deaths in 2019 attributed to just five NCDs: heart disease, stroke, liver cancer, cirrhosis, and stomach cancer [13]. Over the past few decades, the Mongolian government has implemented a number of policies to prevent and control NCDs, such as tobacco control laws and nutrition programs aimed at reducing salt intake [19,20]. However, obesity has received less focus in policy discussions. Enlightening the association between socioeconomic status and obesity is significant for policy. Evidence shows that in high-income countries (HICs), an individual's socioeconomic status is negatively associated with obesity risk in both genders, whereas a positive association is observed in LMIC [21]. Overweight and obesity have been recognized as one of the public health challenges in Mongolia; nonetheless, a distribution of overweight and obesity across socioeconomic status (SES) is not well documented.

This study aims to examine education-related inequality in overweight and obesity among Mongolian adults using data from the Mongolian National Tuberculosis Prevalence Survey which was conducted in 2015 (MNTPS).

## Materials and methods

### Data source

The data for this study were derived from the MNTPS, comprised of 50309 individuals over the age of 15 and was carried out between April 2014 and November 2015. The MNTPS was composed of a nationally representative population-based, cross-sectional survey of households in selected clusters via a multistage random clustered probability sampling design.

Each participant was interviewed using a standardized questionnaire that collected information on demographics, socioeconomic indicators, tuberculosis-related factors, unhealthy behaviors (e.g., smoking and alcohol consumption), and other relevant variables. The main inclusion criteria in the present study were that respondents who are 18 years or older. After applying the inclusion criteria, participants with missing data were excluded, resulting in a final sample of 41,777 participants (69.6% of the eligible sample).

### Ethics statement

The Ethics Review Committee of Kansai Medical University approved the use of a de-identified and anonymized dataset for this study (approval number: 2019278). All data were fully anonymized and de-identified prior to the current analysis. The Ethics Review Committee of Kansai Medical University waived the requirement for informed consent. We accessed the dataset of the current study on March 1, 2022.

### Measuring overweight and obesity

Body mass index (BMI) data were collected during the MNTPS physical examination, and participants' body height and weight were measured by trained personnel. Weight was divided by height squared ($kg/m^2$) to calculate BMI. The previous study found that the World Health Organization (WHO) standard threshold of the BMI cut-off point is appropriate for Mongolians, therefore, we used BMI cut-off points of 25-29.9 $kg/m^2$ for overweight and BMI $\geq$ 30 $kg/m^2$ for obesity [22]. We generated two binary variables, including overweight (if $25 < BMI \leq 29.9$, it is 1; otherwise, it is 0) and obesity (if BMI $\geq$ 30, it is 1; otherwise, it is 0) [23].

### Measurement of socioeconomic status

The concentration index, is a rank-dependent method for measuring socioeconomic-related health inequalities. Different measurements, namely income, consumption, expenditure, wealth index, and education could be used to assess

socioeconomic status [24,25]. In this study, we applied education as a socioeconomic indicator to measure inequality in overweight and obesity. First, in Mongolia, one third of the population are herders whose livelihoods rely on farming and self-production, making it difficult to calculate the real income accurately [26]. Second, education is considered a reliable variable over time [27]. Education offers a more consistent and distinguishable indicator of socioeconomic status and access across different regions [24,25]. Numerous studies have stressed that education can be considered a measure of one's socioeconomic status, and previous research found that the validation of self-reported education level was adequately complete and correct compared to self-reported income variables [28].

Educational levels were classified on the basis of data availability: none or less education (0–4 years), lower secondary education (8 years), upper secondary education (12 years), and tertiary education (16 years or more).

### Independent variables

In this study, in addition to the concentration analysis, the decomposition analysis was conducted to identify potential demographic and socio-economic contributing factors to the concentration index. These contributing factors are regarded as independent variables here.

The independent variables used in this study were age, sex, household income, marital status, employment status, location, smoking status, and alcohol consumption. Age is a categorical variable (18–34, 35–49, 50–64, 65 years or older). Gender is a dummy variable. Marital status was classified as married/living together, divorced/separated, widowed, or single/never married. Employment status is divided into three categories: employed, unemployed, and inactive. Monthly household income is a continuous variable. Location was divided into urban and rural areas. Smoking status was divided into four categories: never smoked, ex-smoker, occasional smoker, and current smoker. Alcohol consumption status was a binary variable (yes/no), with yes indicating that the participant consumed alcoholic beverages more than once in the previous year.

### Measuring inequality

The concentration index (CI) was used to measure socioeconomic inequality in overweight and obese individuals in Mongolia. Since education was used as the ranking variable, education-related inequality was the focus of this analysis. The CI is defined as the covariance between the health variable (in our case, overweight and obesity) and the fractional rank of the education level distribution, as shown in Equation 1:

$$CI = \frac{2}{\mu} cov_w \left( y_{it}, R_i^t \right)$$

(1)

where $i$ is an individual, $y_i$ is the obesity (overweight) variable, $\mu$ is the mean of the obesity (overweight) variable ($y$), and $R_i$ is the individual's fractional rank in the educational distribution. The CI ranges from -1 to +1, with a negative value indicating that obesity (or overweight) is concentrated among less educated individuals, a positive value indicating concentration among more educated individuals, and a zero-value indicating no education-related inequality.

Owing to the binary nature of the health variable (overweight or obesity), the Erreyger's Index (EI) [29] was also used, as shown in Equation 2:

$$E(h) = \frac{4\mu}{(b_n - a_n)} C(h)$$

(2)

where $C(h)$ is the standard CI from Equation (1), $\mu$ is the mean prevalence of obesity (or overweight) in the population, and $b_n$ and $a_n$ indicate the upper and lower bounds of obesity (or overweight), respectively.

## Decomposition analysis

Decomposition analysis was conducted to identify the contributions of various factors to education-related inequalities in overweight and obese individuals. This analysis helps quantify the extent to which each determinant contributes to overall inequality. We used regression-based decomposition for rank-dependent CI. The linear additive model is used for the analysis. The decomposition method for the CI is well documented in the literature and is summarized in elsewhere [24].

Regarding the transformation of overweight and obesity, the EI is equal to the decomposition of the CI multiplied by 4 and *μh*.

$$E = 4[\beta \mu_y C_y + \sum_j \gamma_j \mu_{zj} C_{zj}]$$

(3)

where *μ* represents the mean, *j* is a vector of variables *zj,* γ represents the coefficient of *zj,* and C is the CI [30].

All the statistical analyses were performed with STATA 17.0 (StataCorp LP, Texas, USA).

## Results

### Descriptive statistics

Table 1 presents the sociodemographic characteristics of the study sample by gender. The final sample comprised 41,777 adults, with 60.5% being women. The mean BMI of the population was 26.2 kg/m², with women having a significantly higher mean BMI than men. The prevalence rates of overweight and obesity were 33.4% and 20.7%, respectively, and the obesity was significantly more prevalent among women than men (p < 0.01). Among the participants, 36.5% were aged 18–34 years, whereas 8.5% were aged 65 years or older.

In terms of marital status, 75.1% of the participants were married, with a greater proportion of widowed women than men. Most participants had upper secondary education (48.4%), followed by tertiary education (27.7%), and a small percentage had no or lower education (7.9%). Women were significantly more likely to have higher education levels than men. Alcohol consumption was reported by 67.4% of men and 35.0% of women, whereas smoking rates were notably higher among men, with approximately half being daily smokers compared with 5.6% of women.

Fig 1 illustrates the prevalence of overweight and obesity by education level. Among men, the prevalence of overweight was lowest in the non/low-education group (25.7%) and highest in the tertiary education group (36.3%). Conversely, women with non/low and lower secondary education had a greater prevalence of overweight (32.9%). The obesity rates were lower among women with tertiary education (18.4%) than among those with non/low education (20.6%). For men, obesity rates were higher in the tertiary education group (20.8%) than in the non/low education group (9.5%).

### Social inequalities in overweight and obesity

Table 2 summarizes the EIs for overweight and obesity. The EI for overweight individuals in the total study population was significantly positive, indicating that higher education levels are associated with a higher prevalence of overweight individuals, particularly among men (EI = 0.049). The EI for overweight women was -0.004, which was not statistically significant, leading to its exclusion from further analysis. For obesity, the total EI was -0.003 and not statistically significant. However, the EI by gender revealed that obesity was concentrated among men with higher education (EI = 0.059) and women with lower education levels (EI = -0.047).

### Decomposition analysis

Table 3 presents the decomposition analysis results, showing each determinant's CI, regression coefficient, and the extent of contribution to inequality. For example, unemployment is concentrated among less-educated men (CI = -0.164), which

**Table 1. Descriptive statistics by gender.**

| Variables | Total, % | Men, % | Women, % | P-value[a] |
|---|---|---|---|---|
| | n = 41,777 | n = 16,515 | n = 25,262 | |
| Mean BMI kg/m²; (SD) | 26.2 (5.3) | 25.8 (5.1) | 26.5 (5.4) | <0.01 |
| Weight categories; (%) | | | | |
| Underweight | 1.5 | 1.2 | 1.7 | <0.01 |
| Normal weight | 44.4 | 48.0 | 42.0 | <0.01 |
| Overweight | 33.4 | 33.4 | 33.4 | 0.86 |
| Obese | 20.7 | 17.4 | 22.9 | <0.01 |
| Age group; (%) | | | | |
| 18-34 | 36.5 | 37.0 | 36.2 | 0.07 |
| 35-49 | 31.7 | 31.3 | 31.9 | 0.17 |
| 50-64 | 23.3 | 23.2 | 23.4 | 0.54 |
| 65 < | 8.5 | 8.5 | 8.5 | 0.89 |
| Log income per capita median (SD) | 13.1 (0.87) | 13.1 (0.88) | 13.1 (0.87) | <0.01 |
| Marital status; (%) | | | | |
| Married/living together | 75.1 | 78.0 | 73.2 | <0.01 |
| Divorced/separated | 1.8 | 1.4 | 2.0 | <0.01 |
| Widowed | 6.9 | 2.6 | 9.8 | <0.01 |
| Single/never married | 16.2 | 18.0 | 15.0 | <0.01 |
| Employment status; (%) | | | | |
| Employed | 51.8 | 58.0 | 47.7 | <0.01 |
| Unemployed | 7.0 | 9.1 | 5.6 | <0.01 |
| Inactive | 41.2 | 32.9 | 46.7 | <0.01 |
| Education level; (%) | | | | |
| Non or lower education | 7.9 | 7.7 | 8.1 | 0.19 |
| Lower secondary | 16.0 | 20.6 | 12.9 | <0.01 |
| Upper secondary | 48.4 | 47.7 | 48.9 | 0.01 |
| Tertiary | 27.7 | 24.0 | 30.1 | <0.01 |
| Smoking status; (%) | | | | |
| Never | 71.5 | 40.7 | 91.5 | <0.01 |
| Quitted | 3.1 | 6.3 | 1.1 | <0.01 |
| Occasionally | 3.3 | 5.7 | 1.8 | <0.01 |
| Daily | 22.1 | 47.3 | 5.6 | <0.01 |
| Alcohol consumption in the past year; (%) | | | | |
| No | 52.2 | 32.6 | 65.0 | <0.01 |
| Yes | 47.8 | 67.4 | 35.0 | |
| Residential area; (%) | | | | |
| Urban | 53.1 | 53.3 | 53.1 | 0.67 |
| Rural | 46.9 | 46.7 | 46.9 | |

Abbreviation: % percent, n numbers, BMI body mass index, SD standard deviation.

Weight categories were classified based on BMI according to the WHO classification.

[a]P values were calculated using chi-square test for categorical variables or

Wilcoxon-Rank sum test for ranked continuous variables in order to compare difference between men and women.

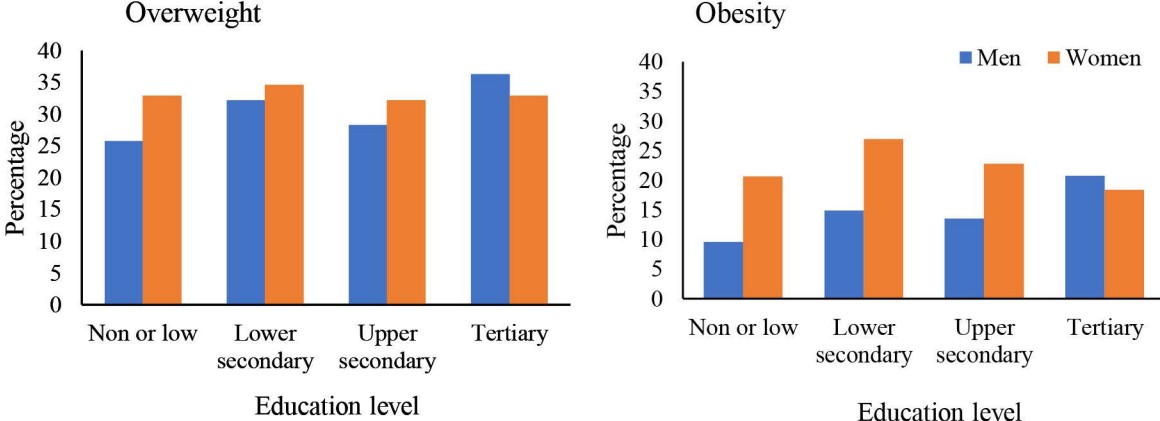

**Fig 1. Overweight and obesity rates among men and women in Mongolia by education level.**

**Table 2. Erreyger's concentration indices of overweight and obesity in Mongolia.**

|  | Overweight | | | Obesity | | |
|---|---|---|---|---|---|---|
|  | Men | Women | Total | Men | Women | Total |
| **EI** | 0.049 | -0.004 | 0.017 | 0.059 | -0.047 | -0.003 |
| **SE** | 0.009 | 0.006 | 0.006 | 0.008 | 0.007 | 0.006 |
| **P-value** | 0.00 | 0.52 | 0.00 | 0.00 | 0.00 | 0.62 |

EI Erreyger's concentration Index; SE Statistical error.

$25 \geq$ BMI 29.9 was considered as an overweight and $\geq 30$ kg/m$^2$ was considered as an obese.

P-value was considered as a statistically significant when below 0.05.

contributes 4% to education-related inequality in obesity. If unemployment was evenly distributed across education levels, education-related inequality in obesity among men would be 4% lower. Moreover, we found that unemployed men are significantly less obese than the employed men.

Conversely, income was concentrated among higher-educated women (CI = 0.012),

which contributes 17% to education-related inequality in obesity among women If income was equally distributed across education groups among women, estimated degree of education-related inequality in obesity would be 17% higher. The results also show that a positive association between income and obesity among women.

Fig 2 illustrates the contributions of individual factors to the CI of education-related inequalities in overweight and obese among adults in Mongolia.

Education-related inequalities in overweight and obesity among men in Mongolia were mainly caused by education, employment status, and income, except the contribution of age.

While obesity was concentrated among lower-educated women which was caused by education, income, alcohol use, and location, except the contribution of age.

## Discussion

This study aimed to examine the education-related inequality in overweight and obesity among adult population of Mongolia using nationally representative data.

Our descriptive analysis revealed that the prevalence of obesity was higher among women (22.9%) than men (17.4%).

**Table 3. Decomposition of concentration indices in overweight and obesity stratified by gender.**

| Variable | CI | | Overweight | | Obesity | | | |
|---|---|---|---|---|---|---|---|---|
| | Women | Men | Men | | Men | | Women | |
| | | | Regression coefficient | Contri-bution % | Regression coefficient | Contri-bution % | Regression coefficient | Contri-bution % |
| Age | | | | | | | | |
| Age 35–49 | -0.023 | -0.089 | **0.075** | -14.2% | **0.095** | -14.9% | **0.145** | 9.7% |
| Age 50–64 | -0.160 | -0.175 | **0.108** | -23.6% | **0.087** | -15.8% | **0.225** | 62.6% |
| Age 65 < | -0.409 | -0.130 | **0.094** | -4.3% | **0.085** | -3.2% | **0.192** | 37.6% |
| Log income per capita | 0.012 | 0.012 | **0.019** | 24.1% | **0.020** | 21.2% | **0.013** | -17.0% |
| Marital status | | | | | | | | |
| Single/never married | 0.042 | 0.023 | **-0.121** | -6.1% | **-0.098** | -4.1% | **-0.077** | 4.7% |
| Divorced/separated | 0.000 | -0.135 | -0.036 | 0.5% | **-0.091** | 1.0% | **-0.056** | 0.0% |
| Widowed | -0.279 | -0.262 | -0.013 | 0.4% | **-0.058** | 1.5% | **-0.031** | -5.7% |
| Employment status | | | | | | | | |
| Unemployed | -0.167 | -0.164 | -0.011 | 1.3% | **-0.039** | 4.0% | -0.013 | -1.1% |
| Inactive | -0.123 | -0.111 | **-0.053** | 14.6% | **-0.023** | 5.3% | -0.007 | -3.2% |
| Education level | | | | | | | | |
| Lower secondary | -0.743 | -0.695 | 0.018 | -17.5% | 0.018 | -14.5% | **0.052** | 40.1% |
| Upper secondary | -0.124 | -0.020 | 0.024 | -2.0% | **0.038** | -2.6% | **0.049** | 25.6% |
| Tertiary | 0.686 | 0.742 | **0.075** | 117.9% | **0.091** | 117.5% | **0.028** | -50.5% |
| Rural areas | -0.137 | -0.163 | 0.005 | -3.0% | 0.002 | -1.2% | **0.015** | 8.0% |
| Alcohol drinking status (No) | 0.141 | 0.043 | **0.026** | 6.2% | 0.008 | 1.5% | **0.024** | -10.3% |
| Smoking status | | | | | | | | |
| Quitted | 0.008 | -0.087 | -0.011 | 0.4% | **0.034** | -1.0% | **0.094** | -0.1% |
| Occasionally | 0.105 | 0.075 | **-0.058** | -2.2% | -0.001 | 0.0% | -0.002 | 0.0% |
| Daily | 0.005 | -0.041 | **-0.048** | 7.4% | **-0.045** | 5.8% | 0.013 | 0.0% |
| Sum | | | 0.049 | 99.88% | 0.059 | 100.4% | -0.047 | 100.47% |
| Residual | | | 0.000 | 0.12% | 0.000 | -0.41% | 0.000 | -0.47% |
| Total CI for obesity | | | **0.049** | 100% | **0.059** | 100% | **-0.047** | 100% |

Significant regression coefficients are in bold, at the significance level of 0.05; CI, concentration index.

Reference groups were: Age 18–34, Married, Employed, None or low education level, Urban area, Alcohol drinking status-yes, smoking status-never

The EI showed that overweight was concentrated among higher educated male individuals.

However, the same was not the case for females. Furthermore, obesity was concentrated among higher educated men and lower educated women.

The higher prevalence of overweight and obesity among men with higher education level aligns with trends observed in other LMICs, such as China, Peru, Iran, Indonesia and Brazil [31–34]. We found that socioeconomic factors such as higher education level, income, and employment status are main drivers for inequalities in overweight and obesity among men. Studies in this context have also highlighted a positive association between higher SES and obesity among men, largely driven by increased access to calorie-dense food, reduced physical activities due to urbanization, and sedentary work pattern. A Peruvian study revealed that higher education levels were linked to lower physical activity, potentially due to sedentary office work [33].

Similarly, a study in Mongolia indicated an increased rate of physical inactivity in urban areas among men due to greater vehicle usage and a sedentary lifestyle, along with extremely high consumption of meat and sugar sweetened beverages [9,18]. In Mongolia, 60% of managerial positions are held by men and the percentage of excessive sedentary

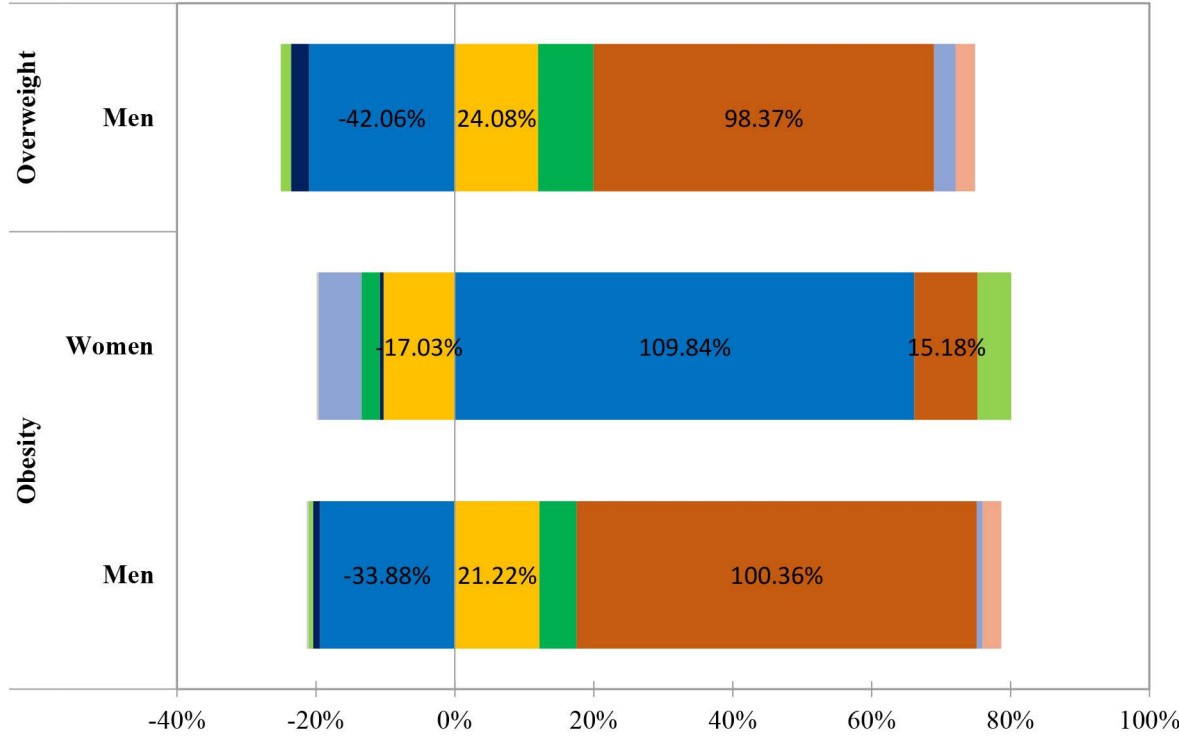

Age ■ Income ■ Marital status ■ Employment ■ Education ■ Location ■ Alcohol ■ Smoking ■ Residual

**Fig 2. Contributions to Erreyger's concentration indices of socioeconomic inequality in overweight and obesity among men and women.**

hours were greater among men than women (44.6% vs 20.9%) [35]. Factors such as greater workload and prolonged sedentary work could limit access to balanced home-prepared meals, leading to increased demand for high-calorie fast food [36].

Yuenan Su et al. found the prevalence of overweight and obesity was higher among high SES participants of the research conducted in Inner Mongolia, due to an increased unhealthy diet with intake of red meat and processed food [34]. Less educated men are more likely to engage in physically demanding labor, contributing to lower obesity rates [37]. In Mongolia, nomadic herding has been a main form of lifestyle for more than a thousand years and a form of livelihood; it requires intensive physical activity from herders, and 25.2% of the total households in Mongolia are nomadic herders [38]. The educational attainment of herders is lower than that of their urban counterparts [39].

Despite the protective effect of higher education on health in other contexts, smoking is a significant factor contributing to inequality in obesity among men in Mongolia, possibly due to lower health literacy [40]. High smoking rates among Mongolian men, coupled with limited health literacy and poor self-management of risk factors, have contributed to the observed findings [40]. Mongolia is burdened by the high prevalence of smoking, with 43.7% among men and 5.0% among women, which is consistent across all education levels [41]. A previous study revealed that knowledge of NCDs and their risk factors was high among women but limited among men across all levels of education. Moreover, a lack of attitudes related to lifestyle and behavioral changes was identified among the Mongolian men [15].

Other factors that might contribute to inequality in overweight and obesity include marital status and age. This may be partly explained by the higher labor market participation rates among younger individuals with higher education levels. Our finding showed that the single men tend to be less obese than the married men which might be explained by that single men may adopt healthier lifestyles or benefit from greater social stability. For example, a most recent systematic review revealed that married individuals have higher odds of developing obesity compared to unmarried individuals [42]. On the other hand, being single in or re-entering marriage markets tends to lose weight due to behavior change and increased physical activities [42].

Conversely, among women, the prevalence of obesity higher in lower socioeconomic groups. This pattern frequently was observed in HICs, and in some LMICs where socioeconomic disadvantage disproportionately impacts women's health. In HICs, a lower SES is associated with reduced physical activity and poorer access to healthy food, whereas higher education levels are linked to healthier lifestyle choices [43–45]. Older age and living in rural areas were significant contributors to obesity in women in Mongolia, which is consistent with studies in other countries such as Indonesia and China [46,47]. In rural areas, the prevalence of obesity among women is noteworthy, with limited education, rural living condition creates barriers to accessing health information, health service, healthy food, and infrastructure contributing to poor dietary patterns and an increased obesity risk [37]. Some studies conducted in Mongolia, revealed that the women in rural areas consume markedly fewer fruits and vegetables during both winter and summer, and their consumption of red meat and refined grains was extremely high compared to urban counterparts [18]. Additionally, challenges in healthcare access and education-related inequalities in healthcare utilization were identified [48].

In this study, older aged women with a lower education level tend to be overweight or obese. This finding could be explained in part by women's physiology and metabolism, pregnancy and mesopause. Additionally, older age and lower SES may have a more significant effect on women in rural areas. For example, older adults in rural regions in Mongolia are vulnerable to poverty, with many unable to have paid work. Their pensions and benefits are often insufficient to sustain a healthy livelihood, and many lack essential information and access to services that could provide needed resources [38]. Domestic and caregiving duties can result in additional time constraints for women in rural areas, which limits their opportunities for physical activity [38]. In Mongolia, the financial ability of older women remains lower than that of older men, partly because of early retirement policies under Mongolian law. According to the Asian Development Bank, women are more likely to be engaged in unpaid labor, both in households and in family-owned businesses such as pastoral husbandry. They are also concentrated in informal jobs and lower-earning sectors such as social services and trade [39]. Moreover, a survey indicated that women are more likely to be physically inactive than men [17]. In addition, marital status, particularly widowhood, contributed negatively to obesity inequality. This finding aligns with a recent meta-analysis indicating that widowhood or divorce is associated with decreased weight and BMI [49]. A possible explanation is financial insecurity due to spousal dependence, which may lead to higher rates of depression and reduced enjoyment of food [50].

Our findings indicate that the need for gender-sensitive, evidence-based policies to address the socioeconomic inequalities driving overweight and obesity in Mongolia should focus on high-risk groups, including men with higher education and income, women with lower education, rural residents, and older individuals. Overall, at the country level, regulation of food marketing, encouraging reformulation of processed foods, and promoting sustainable urban infrastructure that supports physical activities (walkable cities and cycling roads etc.,) are crucial. Also, the development of targeted programs focusing on older women and addressing poverty, pension and healthcare access to improve diet and lifestyle is important. This study is the first to explore education-related socioeconomic inequalities in overweight and obese individuals in Mongolia. However, this study has limitations. We used cross-sectional data from the tuberculosis prevalence survey, which did not allow us to perform a causal analysis and capture the relationship between individuals' income and wealth differences. We used education as a main indicator variable which is a strong determinant of employment and income for adults [51]. However, numerous studies have stressed that education can be considered a measure of one's socioeconomic status. The CI is a descriptive approach that does not provide any supporting evidence that education or

other factors are not determinants of obesity but rather suggests the extent to which education and obesity are associated with one another by comparing low- and high-SES groups. In addition, the decomposition approach does not provide any causal pathway between socioeconomic factors and obesity; however, it reveals additional factors that are correlated with the existing relationship between SES and obesity.

## Conclusions

Education-related inequalities in overweight and obesity exist among Mongolian adults. Future national strategies for tackling obesity should address inequalities in the root social determinants.

## Acknowledgments

The authors thank the MNTPS team and the staff members of the Mongolian National Center for Communicable Diseases.

## Author contributions

**Conceptualization:** Munkhjargal Dorjravdan, Javkhlanbayar Dorjdagva, Katsuyasu Kouda, Enkhjargal Batbaatar, Naranzul Dambaa, Tsolmon Boldoo, Bolor-Erdene Ganbold, Kumiko Ohara, Chikako Nakama, Toshimasa Nishiyama.

**Data curation:** Naranzul Dambaa, Tsolmon Boldoo.

**Formal analysis:** Munkhjargal Dorjravdan, Javkhlanbayar Dorjdagva.

**Investigation:** Munkhjargal Dorjravdan.

**Methodology:** Munkhjargal Dorjravdan, Javkhlanbayar Dorjdagva, Katsuyasu Kouda.

**Project administration:** Munkhjargal Dorjravdan.

**Resources:** Katsuyasu Kouda, Toshimasa Nishiyama.

**Supervision:** Katsuyasu Kouda, Kumiko Ohara, Toshimasa Nishiyama.

**Writing – original draft:** Munkhjargal Dorjravdan, Javkhlanbayar Dorjdagva, Enkhjargal Batbaatar, Bolor-Erdene Ganbold, Kumiko Ohara.

**Writing – review & editing:** Munkhjargal Dorjravdan, Javkhlanbayar Dorjdagva, Katsuyasu Kouda, Enkhjargal Batbaatar, Naranzul Dambaa, Bolor-Erdene Ganbold, Chikako Nakama, Toshimasa Nishiyama.

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
