## [Decision Letter · Decision Letter 0]

PGPH-D-24-02309

Socioeconomic inequality in overweight and obese Mongolian adults: a decomposition approach

Dear Dr. Dorjravdan,

Thank you for submitting your manuscript to PLOS Global Public Health. After careful consideration, we feel that it has merit but does not fully meet PLOS Global Public Health’s publication criteria as it currently stands. Therefore, we invite you to submit a revised version of the manuscript that addresses the points raised during the review process.

We look forward to receiving your revised manuscript.

Kind regards,

Apostolos Davillas

Academic Editor

Journal Requirements:

1. In the online submission form, you indicated that "The data underlying the results presented in the study are available from the National Center for Communicable Diseases of Mongolia.". 

3. Uploaded as supplementary information.

2. Please provide separate figure files in .tif or .eps format.

3. Please provide an Author Summary. This should appear in your manuscript between the Abstract (if applicable) and the Introduction, and should be 150–200 words long. The aim should be to make your findings accessible to a wide audience that includes both scientists and non-scientists. Sample summaries can be found on our website under Submission Guidelines: 

https://journals.plos.org/globalpublichealth/s/submission-guidelines#loc-parts-of-a-submission

Additional Editor Comments (if provided):

Reviewers' comments:

Reviewer's Responses to Questions

**Comments to the Author**

1. Does this manuscript meet PLOS Global Public Health’s publication criteria?

Reviewer #1: Partly

Reviewer #2: Yes

2. Has the statistical analysis been performed appropriately and rigorously?

Reviewer #1: I don't know

Reviewer #2: Yes

3. Have the authors made all data underlying the findings in their manuscript fully available (please refer to the Data Availability Statement at the start of the manuscript PDF file)?

Reviewer #1: No

Reviewer #2: Yes

4. Is the manuscript presented in an intelligible fashion and written in standard English?

Reviewer #1: Yes

Reviewer #2: Yes

Reviewer #1: Report on: “Socioeconomic inequality in overweight and obese Mongolian adults: a decomposition approach”

This study is about the role of inequality in a nutrition-related outcome in Mongolia.

Major comments:

• One of the weaknesses of this study is the claim that the authors analyse socioeconomic inequality when they only use education as a ranking variable. Education is only a component of socioeconomic status (SES). I would recommend better constructing an SES index, which, among other factors, includes education.

• From the title, the authors would focus (and maybe contribute) on decomposition approaches. Rather they do not even explain nor detail which decomposition method/approach they are using. The literature on decomposition methods in health is now very extensive. Each method has its assumptions, pros and cons. Thus, authors should include an entire subsection in the Methods section to provide further details about the decomposition approach followed. Furthermore, authors must argue why they are using such an approach versus other methods.

• The authors should make a strong case for studying the case of Mongolia. Given their current narrative, I cannot see what the study of Mongolia, as a case study, adds to the extensive literature on inequality and nutrition-related outcomes. It looks like the interest may reside in the fact that Mongolia transitioned to a market economy in the 1990s. However, the data used is cross-sectional from April 2014 to 2015. Thus, two recommendations: a) to use data before 1990 and compare before and after the market transition or b) to control for the year of birth in the analysis rather than age.

• Other variables: no discussion nor theory is guiding the inclusion of these variables. There must be at least some justification for the inclusion of each variable. As it is now, it seems more like a convenient exercise than a research-based analysis.

• The authors have done a very superficial literature review. I can identify at least three omitted papers belonging to this literature:

o Davillas, A., & Jones, A. M. (2020). Ex ante inequality of opportunity in health, decomposition and distributional analysis of biomarkers. Journal of Health Economics, 69, 102251. issn: 0167-6296..

o Nie, Peng; Ding, Lanlin; Jones, Andrew M. (2020) : Inequality of Opportunity in Bodyweight among Middle-Aged and Older Chinese: A Distributional Approach, IZA Discussion Papers, No. 13421, Institute of Labor Economics (IZA), Bonn

o Apostolos Davillas, Michaela Benzeval. Alternative measures to BMI: Exploring income-related inequalities in adiposity in Great Britain. Social Science & Medicine, Volume 166,2016. Pages 223-232, https://doi.org/10.1016/j.socscimed.2016.08.032.

o Joan Costa-Font, Joan Gil. What lies behind socio-economic inequalities in obesity in Spain? A decomposition approach. Food Policy. Volume 33, Issue 1,2008,Pages 61-73. https://doi.org/10.1016/j.foodpol.2007.05.005.

Other comments

• Many acronyms never defined, such as STEPS or NCDs

• Line 144-Page 6, clearly define what is the “inclusion criteria”

• Perform sensitivity of results when using another indicator for overweight and obesity, not just BMI

• Why are authors including Marital status in their analysis?

• Is household income self-reported? If yes, how are authors dealing with the great risk of measurement error in this variable?

• In Figure 1, define the education levels, what is the third-level?

• In Figure 2, the colours used for marital status and residual factors are very similar.

• In Figure 2, why is there no decomposition for women overweight?

Reviewer #2: This paper is well-executed and places itself effectively within the related literature. The methodology is sound, and the analysis is well conducted. However, the paper’s contribution to the field is not immediately clear, as it does not sufficiently emphasize how it advances current knowledge or offers new insights. Several areas could benefit from revision and clarification as commented in detail within the report.

**Do you want your identity to be public for this peer review?** For information about this choice, including consent withdrawal, please see our Privacy Policy

Reviewer #1: No

Reviewer #2: No

---

## [Decision Letter · Decision Letter 1]

PGPH-D-24-02309R1

Education-related socioeconomic inequality in overweight and obesity among Mongolian adults

Dear Dr. Dorjravdan,

Thank you for submitting your manuscript to PLOS Global Public Health. After careful consideration, we feel that it has merit but does not fully meet PLOS Global Public Health’s publication criteria as it currently stands. Therefore, we invite you to submit a revised version of the manuscript that addresses the points raised during the review process.

One of the reviewers raised important issues regarding the revised version of your manuscript. Please address all the additional comments provided by the reviewer – of particular relevance, consider offering a full justification for all explanatory variables used in your analysis.

We look forward to receiving your revised manuscript.

Kind regards,

Apostolos Davillas

Academic Editor

Journal Requirements:

Additional Editor Comments (if provided):

Reviewers' comments:

Reviewer's Responses to Questions

**Comments to the Author**

Reviewer #1: All comments have been addressed

Reviewer #2: All comments have been addressed

publication criteria?

Reviewer #1: Partly

Reviewer #2: Yes

3. Has the statistical analysis been performed appropriately and rigorously?

Reviewer #1: I don't know

Reviewer #2: Yes

4. Have the authors made all data underlying the findings in their manuscript fully available (please refer to the Data Availability Statement at the start of the manuscript PDF file)?

Reviewer #1: Yes

Reviewer #2: (No Response)

5. Is the manuscript presented in an intelligible fashion and written in standard English?

Reviewer #1: Yes

Reviewer #2: Yes

Reviewer #1: I appreciate the authors’ responses to my comments.

• Regarding your response to comment 4. I must say, it is tendentious to cite three articles by the same authors as examples of publications where a similar method was used for the analysis. Moreover, the authors heavily justify the lack of discussion about the variables included in their analysis based on the dataset used. There are now several methods based on machine learning designed precisely to avoid cherry-picking. Thus, I strongly recommend to, at least, include a brief justification for the inclusion of the variables mentioned in the “other variables” subsection.

• It is somewhat surprising that age is the main driver of education-related inequality in obesity among women. Please conduct a sensitivity analysis excluding age to test whether nutrition-related inequality is driven by education.

Reviewer #2: You have addressed all comments and I believe the manuscript is now suitable for publication.

**Do you want your identity to be public for this peer review?** For information about this choice, including consent withdrawal, please see our Privacy Policy

Reviewer #1: No

Reviewer #2: No

---

## [Decision Letter · Decision Letter 2]

Education-related socioeconomic inequality in overweight and obesity among Mongolian adults

PGPH-D-24-02309R2

Dr Munkhjargal Dorjravdan,

We are pleased to inform you that your manuscript 'Education-related socioeconomic inequality in overweight and obesity among Mongolian adults' has been provisionally accepted for publication in PLOS Global Public Health.

Best regards,

Apostolos Davillas

Academic Editor

Reviewer Comments (if any, and for reference):

Reviewer's Responses to Questions

**Comments to the Author**

Reviewer #1: All comments have been addressed

publication criteria?

Reviewer #1: Yes

3. Has the statistical analysis been performed appropriately and rigorously?

Reviewer #1: Yes

4. Have the authors made all data underlying the findings in their manuscript fully available (please refer to the Data Availability Statement at the start of the manuscript PDF file)?

Reviewer #1: No

5. Is the manuscript presented in an intelligible fashion and written in standard English?

Reviewer #1: Yes

Reviewer #1: The two additional comments I had were addressed.

**Do you want your identity to be public for this peer review?** For information about this choice, including consent withdrawal, please see our Privacy Policy

Reviewer #1: No
